# The Efficacy of Iron Chelators for Removing Iron from Specific Brain Regions and the Pituitary—Ironing out the Brain

**DOI:** 10.3390/ph12030138

**Published:** 2019-09-17

**Authors:** Robert R. Crichton, Roberta J. Ward, Robert C. Hider

**Affiliations:** 1Faculty of Science, Universite catholique de Louvain, 1348 Louvain-la-Neuve, Belgium; robert.crichton@uclouvain.be; 2Division of Brain Sciences, Imperial College, London W12 ONN, UK; 3Institute of Pharmaceutical Science, King’s College London, London SE1 9NH, UK; robert.hider@kcl.ac.uk

**Keywords:** iron, chelation, neurodegenerative diseases, pituitary, brain

## Abstract

Iron chelation therapy, either subcutaneous or orally administered, has been used successfully in various clinical conditions. The removal of excess iron from various tissues, e.g., the liver spleen, heart, and the pituitary, in beta thalassemia patients, has become an essential therapy to prolong life. More recently, the use of deferiprone to chelate iron from various brain regions in Parkinson’s Disease and Friederich’s Ataxia has yielded encouraging results, although the side effects, in <2% of Parkinson’s Disease(PD) patients, have limited its long-term use. A new class of hydroxpyridinones has recently been synthesised, which showed no adverse effects in preliminary trials. A vital question remaining is whether inflammation may influence chelation efficacy, with a recent study suggesting that high levels of inflammation may diminish the ability of the chelator to bind the excess iron.

## 1. Introduction

Iron is one of the essential elements in the body, the concentration of which is tightly regulated to prevent toxicity. Iron overload, particularly in the liver and spleen, is known to occur in several conditions, such as hereditary hemochromatosis, African iron overload, sickle cell disease, beta thalassemia, sideroblastic anaemia, enzyme deficiency (pyruvate kinase, glucose-6-phosphate dehydrogenase), and rare disorders of proteins involved in iron transport (atransferrinaemia, aceruloplasminemia). In certain conditions, e.g., sickle cell anaemia, beta thalassemia, and myelodysplasia, regular blood transfusions are an essential part of the therapy, thereby contributing to the increased iron stores, as the body is unable to excrete iron to any great extent. However, little focus has been directed at the brain iron concentrations in these conditions. Furthermore, it is also apparent that aging and neurodegenerative diseases, such as Parkinson’s and Alzheimer’s Disease, as well as Friederich’s Ataxia, are associated with an increasing brain iron accumulation [1]. With the advent of several magnetic resonance imaging (MRI) techniques over the past 10 years it is now possible to measure pituitary and brain iron in specific brain regions and to monitor changes in its concentration after iron chelation therapy. Iron deposits are not assayed directly, but their effects on water protons as they diffuse in the magnetically inhomogeneous environment induced by iron deposition are assessed. The scanner transmits energy into the body as microwaves, followed by a waiting period, after which the energy is recalled, to be received by an antennae or coil. This process is known as relaxation and is characterised by relaxation rates R2 and R2* (measured in Hz), which are mathematically inverse of the characteristic relaxation times T2 and T2* (measured in ms).

## 2. Iron chelation

The chelation of excessive amounts of iron from the liver and spleen and, more recently, from the pituitary gland and the brain, has been utilised as a therapeutic approach. In what follows, we will outline the development of the most important therapeutic application of iron chelation therapy to date, namely in the treatment of thalassemia [2,3,4,5] and neurodegenerative diseases [1].

### Iron Chelators in Current Clinical Use

The hexadentate chelator, deferrioxamine (DFO) (Figure 1), a bacterial hydroxamate siderophore, was introduced in the early 1970s, and initially gave poor results because it was not active by oral administration, and had a short half-life (20–30 min). However, the development of continuous subcutaneous infusion of DFO by a portable pump [6] and the establishment of sensible schedules for the optimal use of the pump [7] meant that by the 1990s markedly prolonged cardiac disease-free survival in patents, who faithfully followed the Propper and Pippard regime, could be demonstrated in three independent studies [8,9,10]. However, sadly, half of the patients, who could not or would not comply, developed cardiac failure or arrhythmia much more rapidly [11]. What was urgently required was a chelator which could be used more easily, thereby improving compliance, and which was both effective and orally active.

Deferiprone (DFP) a small, lipophilic bidentate chelator of the 3-hydroxypyridin-4-one family (Figure 1), was introduced into clinical practice in the 1980s [12]. Although it is orally active, its half-life is 3–4 h, which means that it must be administered three times a day at doses of 75 mg/kg/day to maintain sufficient negative iron balance equivalent to 50 mg/kg/day of DFO [13,14]. DFP has good bioavailability, but its clearance is accelerated by rapid biotransformation: approximately 85% of the drug is metabolised to a nonchelating 3-*O*-glucuronide conjugate [15]. The most important side effects of deferiprone are agranulocytosis and milder forms of neutropenia, which require appropriate monitoring [16]. However, DFP enters cells and can access intracellular chelateable iron more readily than DFO [17]. DFP possesses cardioprotective effects [18], and a multi-centre prospective comparison showed that a combined DFP + DFO regimen was more effective in removing cardiac iron than DFO, and was superior in clearing hepatic iron than either DFO or DFP monotherapy [19].

In the years that followed the introduction of DFP, the search for other orally active chelators intensified [20]. The microbial tridentate chelator, desferrithiocin, discovered in 1980, was shown to be orally active and very effective in mobilising hepatic iron in a rat model with hepatic iron loading induced by a ferrocene derivative [21], and in iron-loaded monkeys [22,23]. However, ferrithiocin itself proved to be toxic in animals [24], and although many desferrithiocin derivatives were tested and found to be more effective than desferrithiocin, toxicity remained associated with this class of chelators [25]. The clinical development of Deferasirox (DFX) (ICL670) represents an investment, the magnitude of which has no precedent in the history of chelator research. In the search for a safe tridentate chelator, a completely new chemical class of iron chelators, the bis-hydroxyphenyltriazoles, was discovered. More than forty derivatives of the triazole series were synthesised at Novartis, and evaluated, together with more than 700 chelators from other chemical classes. The tridentate chelator 4-[(3,5-Bis-(2-hydroxyphenyl)-1,2,4) triazol-1-yl]-benzoic acid (ICL670, Figure 1) emerged as the chelator which best combined high oral potency and tolerability in animals [20]. DFX has a plasma half-life of 12–16 h, which allows it to be taken once per day, effectively eliminating non-transferrin bound iron (NTBI) from the circulation [26]. In clinical studies, DFX has proven to be as effective as DFO at doses of 20–30 mg/kg [27], and in an extensive study involving 1744 patients, it was shown that fixed starting doses of DFX, based on transfusional iron intake, with dose titration guided by serum ferritin trends and safety markers, provide clinically acceptable chelation in patients with transfusional hemosiderosis from various types of anaemia [28]. Side effects are minimal, although the use of DFX is associated with rash formation and renal toxicity [29].

## 3. Thalassemia, Sickle-Cell Anaemia, and Haemoglobinopathies

Thalassemia, sickle-cell disease, and other hereditary disorders of haemoglobin biosynthesis are the most prevalent monogenic diseases worldwide [5]. Thalassemia results from an inherited defect in the rate of synthesis of one of the two polypeptide chains of haemoglobin, resulting in imbalance in the α,β-globin chain ratio, ineffective erythropoiesis, chronic haemolytic anaemia, and increased intestinal iron absorption. In the 1960s and 1970s, the only effective treatment for thalassemia was blood transfusion. Since humans cannot increase their iron excretion to compensate for iron loading, the transfusional iron overload, aggravated by increased intestinal iron absorption, results in massive and progressive iron overload. Each unit of transfused blood contains approximately 200 mg of iron, and since the mean transfusional loading in thalassemia major is 0.4 mg/kg/day, it comes as no surprise that most patients faced certain death before the age of 20 years from heart failure. However, the incidence of heart failure was not directly related to the cardiac iron overload, but to the level of iron loading in the liver. The explanation for this lies in the demonstration by Chaim Hershko [30] of the presence in the serum of thalassemic patients of toxic non-transferrin-bound iron (NTBI) which was responsible for the cardiac damage. When the liver iron load attains a critical level, iron is released into the circulation, and once its concentration exceeds the binding capacity of the transferrin pool, NTBI is formed. NTBI is the main source of iron accumulation in iron loading in both thalassemia and hereditary hemochromatosis, notably in the heart, pancreas, and liver [31], as well as in the brain. The influx of NTBI seems to be mediated by the ZIP14 transporter in the liver [32,33], whereas in cardiomyocytes, two calcium channels transport Fe^2+^ with an affinity similar to that of Ca^2+^ [34,35], and their expression is insensitive to cardiac iron overload.

### 3.1. Thalassemia-Iron Accumulation in the Pituitary

Clinical hypogonadism is common in thalassemia patients (83%) with detectable cardiac iron and is independently associated with pituitary iron deposition and gland shrinkage [36]. The pituitary gland maintains its anatomical and functional connections with the brain, yet sits outside the blood–brain barrier. Iron deposition in the anterior pituitary continues to pose a serious problem in homozygous beta thalassemia patients, particularly in terms of gonadal function. Magnetic resonance imaging (MRI) measurements are able to estimate the amount of iron in the pituitary, which can then be correlated with gonadal function. The anterior pituitary gland is particularly sensitive to free radicals produced by oxidative stress, such that exposure to these radicals injures the gland. MRI has shown that even a modest increase in iron deposition in the anterior pituitary can lead to dysfunction, such as abnormalities in the hypothalamic–pituitary growth hormone axis and growth hormone neurosecretory dysfunction, and low insulin-like growth-factor-1 levels. Abnormalities in the hypothalamic–pituitary–gonad axis will induce lower follicle stimulating hormone (FSH) and luteinizing hormone (LH) secretion, low LH/FSH response, and low sex steroid secretion from the gonads, i.e., testosterone (reviewed [37]). The spleen plays an important role in destroying abnormal red cells, sequesters 30–40% of the circulating platelet pool, and plays a role in the regulation of plasma volume. In thalassemic patients, the spleen may become enlarged, which can result in an increased propensity to recurrent infection, such that a splenectomy is required. Following splenectomy, the pituitary iron loading increases [38]. In early studies of four thalassemia patients, (Hb-E thalassemia and thalassemia major), the pituitary iron content, as assessed by MRI, was increased in all four patients compared to controls [39]. A subsequent MRI study of 84 patients with β-thalassemia established a correlation between pituitary iron overload, (evaluated by T2*) and both hepatic and cardiac iron load. In addition, pituitary MRI values correlated with serum ferritin and patient age, but not with height of the pituitary [38]. A study of 30 children and young adult thalassemic patients (13 females and 17 male patients) confirmed iron accumulation in the liver, myocardium, and the pituitary, a linear regression between pituitary iron and age in patients > 14 y, while MRI values between the pituitary and liver, and liver and myocardia, were only moderately correlated, (r = 0.34 and 0.42, respectively). However, no correlation was evident between pituitary and myocardial MRI results, which were of interest, since it might be considered that NTBI would be taken up by both of these organs [40]. It is clear that a therapeutic approach based on the use of iron chelators offers a solution to many of these dilemmas, associated with systemic iron overload, and their development has resulted in innumerable lives being saved, and, just as importantly, has led to an enormous improvement in the quality of life of thalassemia patients all over the world [5]. Investigations have now been extended to monitor whether excessive amounts of iron in specific brain regions and the pituitary can be removed by iron chelators.

### 3.2. Iron Chelation from the Pituitary Gland

Iron chelators, such as deferoxamine, deferiprone, or deferasirox, could be used alone or in combination to induce negative iron balance and reverse hypogonadism and endocrine complications in severely iron-overloaded thalassemic patients. Therefore, it is of interest to ascertain whether iron chelation can reduce iron overload in the pituitary as well as the liver, heart, and spleen. Berkovitch et al. [41] investigated 33 patients >15 years old with transfusion-dependent homozygous beta thalassemia, all of whom had received DFO. Anterior pituitary function (gonadotrophin releasing hormone, GnRH, stimulating test) correlated well with the MRI results for iron deposition in the anterior pituitary, although there was no correlation between the MRI measurements, the GnRH stimulation test, and the clinical status of the patients. Interestingly, 28 out of the 33 patients achieved normal puberty. The deficit in endocrine function in thalassemic patients was confirmed in a study of 78 male thalassemic patients (4–11 years) treated with frequent transfusions and long term DFO therapy, who were compared with 30 age- and sex-matched control children. The prevalence of hypogonadotropic hypogonadism in the thalassemic patients was 76.2%, with a significant increase in serum ferritin (×7), and significantly reduced the serum level of cortisol, growth hormone, stimulating hormones and testosterone [42].

Using the new oral chelator DFX for a two-year period, Wood et al. [43] investigated the pituitary iron content and volume in 31 chronically-transfused patients (28 thalassemia major and three Blackfan Diamond). MRI measurements were taken at the baseline, one, and two years. Twenty-six patients completed the study, 10 were prepubertal, 12 had achieved normal puberty, and four were hypogonadal at base line. Interestingly, the decreased pituitary volume at baseline returned to normal values after two years of DFX therapy, while pituitary iron content as assayed by R2* showed no net change. This latter result was of interest, as it was expected that with increasing age, pituitary iron would increase in parallel (predicted to be a 0.4 increase) [36].

Such studies indicate that iron chelators are able to reduce pituitary iron content as well as improve its function. It is clearly of interest to know whether brain regions are vulnerable to increased iron deposition in β-thalassemic patients. There have been three main studies which investigated other regions of the brain by quantitative MRI. In the studies of Duprez et al. [44] and Hasiloglu et al. [45], various MRI methodologies were utilised to identify an increase in iron in the choroid plexus of β-thalassemic patients. Qiu et al. [46] investigated the distribution of iron in different brain regions of 31 thalassemic patients, aged 25.3 ± 5.9 years, and 33 age matched healthy volunteers. Of the 31 patients, 27 exhibited abnormal iron deposition in one of the brain regions investigated by quantitative susceptibility, showing significantly lower susceptibility values in the globus pallidus and substantia nigra and significantly higher susceptibility in the red nucleus and choroid plexus. In the control subjects, there was a positive age effect on susceptibility value in the putamen, dentate nucleus, substantia nigra, and red nucleus. However, more information is needed with respect to the liver and heart iron deposition and the extent of chelation in each of the thalassemic patients for complete interpretation of these results. Surprisingly, there have been no studies on iron distribution in specific brain regions of thalassemic patients, before or after iron chelation. Whether continuous iron chelation in these patients will preclude such iron accumulation is unknown, nor whether there is preferential iron accumulation in specific brain regions. Further studies are clearly required.

## 4. Parkinson’s Disease—Iron Accumulation in the Substantia Nigra

Parkinson’s disease is the second most common form of motor system degeneration, and is characterised by a progressive loss of dopaminergic neurons in the substantia nigra pars compacta in the ventral midbrain. There is an increased accumulation of iron in the substantia nigra (SN) [47], with a smaller accumulation of iron in other brain regions, such as the red nuclei, globus pallidus, and the cortex of Parkinson’s Disease (PD) patients. As the severity of the disease increases the total iron content increases in the SN, which correlates with motor disability [48] and microgliosis [49]. Semi-quantitative histochemical studies of the SN show that iron deposits are present within the neurons and glia of the substantia nigra, putamen, and globus pallidus, with an increase in ferritin-loaded microglia cells in the substantia nigra [50]. The increase in iron in the SN of PD patients is associated with increased ferritin and neuromelanin iron loading [50,51], as well as increased expression of divalent metal transporter 1, which may contribute to PD pathogenesis via its capacity to transport ferrous iron [52]. Intense microgliosis occurs around extra neuronal neuromelanin (released by dying neurons) in the substantia nigra of patients with Parkinson’s disease [53,54], which could be an important factor in inducing iron accumulation within the microglia [49].

Preliminary reports suggested that identification of neuromelanin, NM, (rather than iron) by MRI might be feasible in patients with Parkinson’s disease [55,56]. Indeed, the loss of neuromelanin in the locus coeruleus and substantia nigra, together with the associated loss in the ability to sequester iron, might be a characteristic sign of Parkinson’s disease [56]. Recent studies have been able to discriminate neuromelanin in PD brains. For example, three-dimensional (3D) neuromelanin-sensitive ^31^MRI [57] showed that signal densities and contrast ratios were significantly lower in the SN of PD patients compared to controls. Another study of 13 late stage PD patients (LSPD) and 12 de novo PD patients (2–5 years duration) showed that the signal was significantly decreased in LSPD compared to de novo PD, while in the lateral SN region, a decrease in the contrast ratios was detected in all PD groups compared to controls [58]. Interestingly, in this study, the NM signal area was significantly correlated with Hoehn Yahr Stage and Movement Disorder Society Unified Parkinson’s Disease Rating Scale (MDS-UPDRS) part II, while a weak correlation was found with MDS-UPDRS part III. Such results may indicate that measurement of the neuromelanin content in the brain by MRI techniques could be an important diagnostic approach.

### Chelation of Iron in the Brains of Parkinson’s Disease Patients

Chelation of iron from the SN of PD brains was proposed many years ago. In our initial animal studies over 15 years ago [59,60], we showed that iron chelators (DFO and DFP) already in clinical use for the treatment of iron loading systemic diseases were able to cross the blood brain barrier (BBB), reduce the iron content in various brain regions, and induce neuroprotection in an animal model of PD. In 2010, Kwiatkowski et al. [61] reported the beneficial use of DFP in one PD patient. After one year’s administration of DFP, at 30 mg/kg/day, there was an improvement in the UPDRS score and a decrease in iron accumulation in the bilateral dentate nuclei, as well as the SN [61]. It is of interest that the time period to observe a beneficial action was approximately one year. Subsequent clinical trials in 2011 and 2013 confirmed that it was necessary to administer DFP for at least nine months at a comparable dose to observe a beneficial effect, reflected by changes in the UPDRS score. Both studies [62,63] identified the beneficial effects of DFP. In the study of Devos et al. [62], R2* sequences were used to assess iron content in the SN, while UPDRS scores were acquired at various intervals to assess clinical parameters, and serum ferritin was assayed as a marker of iron stores. After six to nine months, decreases in SN iron content were quantitated by MRI, and there were improvements in both motor and UPDRS motor scores. In the second study, of six months duration [63], decreases in SN iron content were assayed by MRI T2* in patients receiving 30 mg/kg/day, and there were indications of an improvement in UPDRS. Interestingly, PD patients exhibiting high inflammatory markers in the blood, e.g., IL-6, did not respond well to iron chelation, leading to the suggestion that chelatable iron was not freely available for chelation. There was no evidence that the chelation therapy had an adverse effect on haematological parameters. Interestingly, in the study of Bastida et al. [63], the chelation of iron from other brain regions was also assessed. After three months, the caudate nucleus showed decreased iron content, while the dentate nucleus showed an iron decrease at six months. At this time there was also evidence for a decrease in substantia nigra iron content, which took up to nine months in the Davos study. Clearly, this indicates that there is a selective removal of iron from different brain regions. The major disadvantage of the use of DFP was the incidence of neutropenia and agranulocytosis, which occurred in 8% of the patients. This side effect resolved rapidly with cessation of the oral therapy. Such a side effect required that all PD patients entering the clinical trial undertook a weekly white cell count. These positive results from the clinical trials have confirmed that iron chelation therapy has potential as a therapeutic option for the treatment of PD; if the side effect could be eradicated, it would become a much more universal treatment. In addition, it remains unclear whether the long-term use of such chelators might alter oligodendrocyte function, since they have a large requirement for iron. It is therefore suggested that clinical studies that use iron chelation therapy for diseases associated with iron deposition should monitor white matter (WM) changes. A number of these diseases already report apparent hypomyelination, and further decline of WM integrity will impact the functional outcomes, since myelin integrity impacts movement and cognition. Currently, a multi-centre phase 2 clinical trial involving 300 patients from all over Europe is in place, centered at the University of Lille, France.

Over the past decade, work has been directed towards the design of hydroxypyridinones, which lack an effect on white blood count in rodents and primates. A lead compound, CN128 (Figure 1), has been identified, which possesses all of the therapeutic properties of DFP, including penetration of the blood–brain barrier, but lacks the side effects of induction of neutropenia and agranulocytosis in primates after prolonged exposure (nine months) [64]. CN128 is currently undergoing tests in a range of Parkinson’s disease models.

## 5. Alzheimer’s Disease

Alzheimer’s disease (AD) is a fatal age-related neurodegenerative disease which results in cognitive decline, memory loss, and psychosis. Clinically, AD is characterised by progressive dementia. Initial symptoms are short-term memory loss, which is followed by extensive neuronal loss in the hippocampus and selected cortical and subcortical area. There is abnormal protein processing, with accumulation of the β-amyloid peptide, which is deposited extracellularly and manifests itself as neuritic amyloid plaques, and of the hyperphosphorylated tau protein, in the form of neurofibrillary tangles. Both of these abnormal protein aggregates are definitive markers of the disease in post mortem material.

There is considerable evidence that there is defective homeostasis of iron in the AD brain. Increases in the iron content of AD have been reported for over 50 years. In a recent paper, which made a comprehensive systematic meta-analysis and review of over 2556 publications, 43 eligible studies were identified where the iron content in AD serum, cerebrospinal fluid (CSF), or brain tissue had been studied. In nineteen studies, the iron content in 12 selected brain regions was analyzed by separated meta-analysis, and it was concluded that eight specific brain regions (the temporal, parietal, and frontal lobes) had higher iron concentrations, which correlated with the clinical diagnosis of AD [65]. A recent laser ablation inductively coupled mass spectroscopy study also identified increases in iron in the frontal cortex [66].

### Alzheimer’s Disease—Iron Chelation from the Cortex and Hippocampus

Current therapies for Alzheimer disease (AD), such as the acetylcholinesterase inhibitors and the NMDA receptor inhibitors, may provide moderate symptomatic delay at various stages of the disease, but do not arrest the disease progression or induce meaningful remission. The confirmation of altered iron homeostasis in the brain of AD patients has opened the possibility of using iron chelators as a new therapeutic approach. Various animal models of AD have been utilised to investigate the therapeutic action of iron chelators. For example, the iron chelator (-) epigallocatechin-3-gallate and M-30 (Figure 1) reduced amyloid precursor protein (APP) expression in cultured cells [67,68]. Although it is known that there is an increase in iron in various brain regions in AD patients, there has been no clinical evidence which supports the use of chelating agents as an adjunctive treatment for AD. Tea polyphenols and curcumin (Figure 1) have been advocated as metal chelators for the treatment of AD, although the efficacy of such natural products awaits further investigation.

In 1991, McLachlan showed that DFO significantly reduced the behavioral and cognitive decline of AD patients [69]. However, since this ground-breaking study it is only now that further studies are being undertaken to investigate the therapeutic efficacy of iron chelators in AD. Currently, a phase 2, randomised, placebo-controlled, multi-centre study to investigate the safety and efficacy of DFP has started recruiting. Approximately 171 participants with Prodromal Alzheimer’s disease and mild Alzheimer’s disease will be recruited for the study. Controlled release DFP will be used in an attempt to prevent the unwanted side effects of neutropenia and agranulocytosis. The aim of this study is to ascertain whether DFP (15 mg/kg BID orally) slows cognitive decline in Alzheimer’s patients. MRI will ascertain the iron content of brain regions during the period of the study (Neuroscience Trials Australia—Clinical Trial Deferiprone to Delay orallyDementia).

Other compounds are in development for the treatment of AD, which have multitargeting properties. They include a series of (3-hydroxy-4-pyridinone)-benzofuran hybrids [70] which mimic donepezil, an inhibitor of acetylcholinesterase, and possess additional properties, such as metal chelation, radical scavenging, and the inhibition of amyloid peptide aggregation. In addition, a series of thiosemicarbazones, (pyridoxal 4-N-(1-benzylpiperidin-4-yl) thiosemicarbazone compounds exhibit very low anti-proliferative activity, substantial iron chelation efficacy, inhibition of copper-mediated amyloid-β aggregation, inhibition of oxidative stress, moderate acetylcholinesterase inhibitory activity, and autophagic induction [71]. Although such compounds show neuroprotective effects in vitro, it remains to be investigated whether they have a beneficial effect in either animal models of AD or AD patients.

## 6. Friederich’s Ataxia

Friedreich’s ataxia (FRDA) is one of a number of neurological disorders caused by the anomalous expansion of unstable nucleotide repeats in which, unlike Huntington’s disease, the nucleotide expansion occurs in the non-coding part of the gene (the introns). FRDA is characterised by a progressive degeneration of large sensory neurons and cardiomyopathies [72]. Although rare, FRDA is the most frequent inherited ataxia, with an estimated prevalence of two to four people in 100,000 individuals. Most patients carry homozygous GAA expansions in the first intron of the frataxin gene on chromosome 9. Whereas the critical pathologic triplet repeat threshold is 66 repeats, the expansion can be as many as 890 GAA repeats. This results in the partial silencing of frataxin, a small mitochondrial protein which plays an essential role in iron–sulfur cluster (ISC) biogenesis, an essential metabolic pathway found in all organisms [73,74]. Frataxin interacts directly with the two central components of the ISC biogenesis machine, the NFS1/IscU complex [75]. Correct ISC synthesis in mitochondria is closely linked to cellular iron homeostasis [76], and a lack of frataxin therefore causes dysregulation of iron metabolism. As a consequence, failure to assemble mitochondrial Fe–S proteins results in increased cellular iron acquisition, mitochondrial iron overload [77], and mitochondrial iron deposits in some FRDA patients [78].

### Iron Chelation from the Dentate Nucleus in Friederich’s Ataxia Patients

In one clinical trial, deferiprone (10–15 mg/kg) was administered twice daily to FRDA patients in a small clinical trial over a six-month period to nine adolescent patients with no overt cardiomyopathy. Brain iron was reduced significantly in dentate nuclei. The chelator treatment caused no apparent hematologic or neurologic side effects, while reducing neuropathy and ataxic gait in the youngest patients [79]. In a second clinical trial, DFP (10 mg/kg) was administered in combination with idebenone (20 mg/kg) to 20 FRDA patients. No significant differences were observed in the total international cooperative ataxia rating score (ICARS) scores when comparing baseline status and the end of the study in the whole group of patients. Echocardiography data showed a significant reduction of the interventricular septum thickness and in the left ventricular mass index. After 11 months of treatment, iron was reduced in the dentate nuclei. However, there was a worsening of posture and gait compared to baseline [80].

## 7. Conclusions

The current longevity of the population will result in increasing iron loading in various brain regions, although this is not normally associated with toxicity. Exactly why and how the iron which accumulates in specific brain regions in neurodegenerative diseases is highly toxic is unclear. However, new studies of activated microglia indicate that these cells may, in part, be the source of the iron loading. Over the past 50 years chelation therapy has progressed from subcutaneous administration to oral administration, and it is hoped that toxicity that has been associated with the oral chelators will be eliminated with new formulation of the hydroxypyridinones. 

## Figures and Tables

**Figure 1 pharmaceuticals-12-00138-f001:**
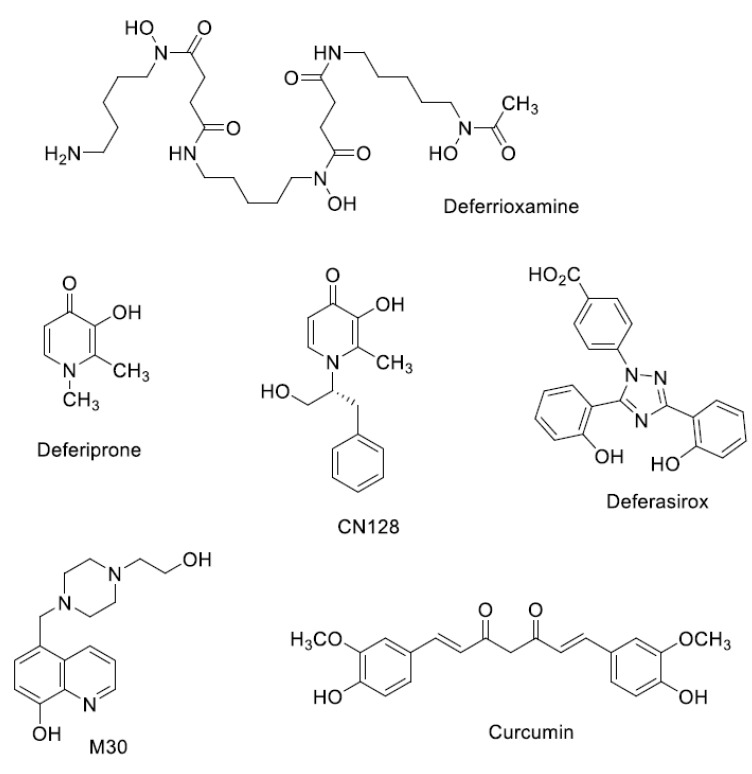
Chemical structures of iron chelators.

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
