# Peer review of "The Efficacy of Iron Chelators for Removing Iron from Specific Brain Regions and the Pituitary—Ironing out the Brain"

_pharmaceuticals, 2019, doi:10.3390/ph12030138_

Round 1

Reviewer 1 Report

Review of the manuscript: In this manuscript, the authors review the importance of iron chelators in the therapy of organ-overload non-transferrin bound iron (NTBI) related diseases, such as  hemochromatosis and neurodegenerative diseases. The cited overload iron chelators were either siderophores from microbes or derivatives. To support their point, the authors selected various  case studies that focus on the benefit of using those chelators in the treatment regimen of genetic disorders or diseases directly or indirectly linked to iron overload; causing toxicity in certain organs, such as spleen, heart, pituitary gland and other regions of the brain. The reason why this NBTI is toxic and how this toxicity occur is reported to be unknown. Nevertheless, the authors report the use of iron chelators deferrioxamine, deferiprone and desferasirox in decreasing of the level of NTBI in multiple cases. Accumulation of NBTI in the anterior of the pituitary gland is one of the characteristic depicted in thalassemia patients. NTBI chelators, deferrioxamine, deferiprone and desferasirox are often used solely or in combination to remove labile iron. Among the iron chelators, deferrioxamine and deferiprone are reported to cross the blood brain barrier (BBB) which allows them to be used as NBTI sequester in parts of the brain as well as the nervous system. Studies cited in this manuscript state that decreases are observed in iron accumulation in the bilateral dentate nuclei and that there are improvements of the Unified Parkinson's Disease Rating scale (UPDRS) score after the administration of 30mg/kg/day of deferiprone in Parkinson's Disease patients. Ultimately, a clinical trial involving the utilization of deferiprone to a dose of 10-15mg/kg   is reported to decrease the brain iron significantly in dentate nuclei. The authors propose a future formulation that involves the combination of neurodegenerative drugs and the iron chelator for better results. Even though the case studies reported here show that NTBI might be toxic and might play a role in worsening the situation of patients with the aforementioned conditions, it is difficult to support such affirmation because of the lack of quantitative data and figures. Another flaw of this manuscript is that the authors did not review whether there is a threshold of the NTBI in the studied organs that is considered to be a toxic concentration. Such information is important insofar as it will help to establish the efficacy of the chelators based on the maximum response but not based on comparison with deferrioxamine. To me, the authors fail to state the efficacy of any of the chelators because of absence of quantitative data and figures that clearly show the maximum response for the dose used. Furthermore, in any part of the manuscript was stated the fate of the complex iron-Chelator, whether it is excreted or metabolized otherwise, it might turn into a source of toxicity upon accumulation.

Upon a better job on presenting more quantitative data, such as figures and tables that clearly show the potency of these NTBI chelators on removing these NTBI, better job on reviewing the concentration of NTBI that is considered toxic for the cited organs and establishment of efficacy of the chelators, this review could be a valuable contribution to the field. I would consider rereviewing the article after significant revision.

Specific Comments

1. The manuscript reads like a first draft. Many sentences are incomplete and/or with incorrect punctuation.

Examples:

P3L81- “20-30mg.kg”

P4L120- “i.e. testorone (reviewed by [37].”

P5L162- place a period after “no net change”.

2. There are numerous parts where the font and font size are inconsistent.

An example of this is found on P3L60.

3. The authors do not seem to be aware of the very serious side effects of deferasirox. There have been claims about fatalities from use of this chelator. The authors need to modify their claim on page 3 lines 84-85 to reflect the additional consequences of use of this chelator.

4. I recommend the inclusion of a figure of the brain, highlighting the regions that are mentioned throughout the manuscript.

5. The authors rely on MRI data as quantitative analysis of iron levels. I think the authors should be careful when discussing these data because they refer to MRI results without specifically saying so and this may confuse readers unfamiliar with MRI.

An example is P5L162- “assayed by R2* showed no net change”

6. Please revise P4L128 to provide more clarity.

7. More clarity is needed regarding the white matter that the authors allude to as mentioned on P6.

8. While the authors specifically state that they will focus on the role of iron overload in these diseases, they should acknowledge that the overload of other metals is also relevant. Without this acknowledgement, someone new to this field might be misled.

9. Revise P7L299-300 for clarity.

Conclusion

This section felt very unsubstantial and requires significant revision. The authors did not do any conclusion on the efficacy of these NTBI chelators. Which chelator and which dose is reported to be optimal in removing NTBI in the different organs as stated in the body of the text? The very last sentence is particularly disappointing considering that they discussed a few different types of chelators but only mention one chelator.

Author Response

We are somewhat irritated by the negative and abusive comments of Reviewer 1 to this invited article. This reviewer appears to be labouring under a totally false impression. We are not addressing the chelation of NTBI, the form of iron found in conditions where transferrin saturation is high and this particularly toxic form of iron is found in extracellular fluids such as plasma. After briefly reviewing the development of the three iron chelators which are currently approved for the treatment of secondary iron overload, we focus our attention on the use of these chelators to improve sexual development in thalassaemia patients, and then in the treatment of neurodegenerative disorders characterised by the accumulation of intracellular iron at levels which do not result in the presence of NTBI. Our concern is with the removal of iron from specific brain regions as assessed by MRI (in the absence of any other way of evaluating brain iron) in two phase II clinical trials on PD patients. The studies on FA likewise did not involve chelation of NTBI, while in the studies reported on iron removal from the pituitary, again only tissue iron levels assessed by MRI were involved.

For this reason, we consider that the general remarks made by this reviewer reveal a total lack of understanding of what the studies were endeavouring to establish, and indeed of what constitutes a clinical trial.

Specific comments.

This is not a first draft- we have reviewed the paper thoroughly. The small number of missing ‘points ‘ have been corrected.

A slash, 20-30mg/kg, a bracket (Reviewed [37]) and a point, no net change. have been added. The reviewer clearly cannot spell testosterone!!

Our text do not show the changes in font and we must conclude that during the transfer of our manuscript onto the formatting for the book these changes have occurred. This remark about the toxicity of desferiserox are not valid. All iron chelators show toxic side effects such that patients are very carefully monitered, Deferiprone may induce reduced granulocyte counts, desferrioxamine may cause allergic reactions and blurred vision. Desferisirox has been shown to induce nephrotoxicity which can be prevented by less aggressive iron depletion strategy ( see Diaz-Garcia et al., Nat Rev Nephrol. 10, 574-586). We do actually state in the text that deferasirox ‘should be used with dose titration guided by serum ferritin trends and safety markers’ We do not feel that a figure is needed as many regions of the brain are adversely affected by iron loading in neurodegenerative diseases. MRI is the only method currently available for non-invasive semi quantitaive assessment of brain iron in specific brain regions. We have now described the technique in the first part of the document and explained the terminology of T2, T2* and R2 R2*. The sentence is clear to us Not necessary as this is merely pointing out that oligodendrocytes need to be monitored carefully during iron chelation therapy. If this reviewer knows of studies where the distribution of other metals have been quantitated non-invasively in patients with various neurodegenerative diseases we would like to know the reference. However since the title of the book is Iron as Therapeutic Targets in Human Diseases it seems strange that this reviewer thinks that we should start discussing other metals than iron! Again we feel that this sentence is clear.

This reviewer should realise that only deferiprone has been used extensively in clinical trials of neurodegenerative diseases.

Reviewer 2 Report

The manuscript Robert R. Crichton’s et al. “The Efficacy of Iron Chelators from the Specific Brain Regions and the Pituitary-Ironing Out of the Brain” is an up-to-date review of successful iron chelation therapy of iron overload disorders. The authors consider two groups of diseases. The first group includes diseases associated with inborn errors in the haemoglobin biosynthesis (mainly monogenic hereditary pathologies), in which iron accumulation occurs in various cellular compartments: in the cytosol (thalassemia) or mitochondria (sideroblastic anemia) of the visceral organs. The second group is neurodegenerative diseases characterized by iron deposits in different parts of the brain due to various, and not always obvious, causes. The following questions arise.

Is there a relationship between the iron accumulation cause, the deposit localizations and the effectiveness of specific iron chelators. If there is such a connection, it would be useful to discuss it. Are there any features of chelation therapy for cases that develop due to copper dyshomeostasis?

Minor points:

Please, add the references in Introduction for each maxima. It is necessary to put spaces correctly: not 12-16h, but 12-16 h; not 20-30mg/kg, but 20-30 mg/kg; not ii)Iron chelation from the Pituitary Gland, but ii) Iron chelation from the Pituitary Gland, and so on throughout the text, including references. Lines 110, 111, 121, and others: are the allocations in italics, underline, or another font justified? Lane 214: remove comma after “ago” Lane 343: Funding: do you have no funds? Bibliography needs to be checked for accuracy.

Author Response

In our opinion references are not normally included in the abstract

Spaces have been added as requested

We do not understand why the format was changing as in our text it is OK.

We have checked the references.

Round 2

Reviewer 1 Report

P1L33-35.- In the revise manuscript, the authors did not review whether there is a threshold for a toxic iron concentration as recommended but mention that their effects on water protons as they diffuse in the magnetically inhomogenous environment induced by iron deposition are assessed. The question is assessed by what technic? I am guessing that they were probably thinking MRI, as they describe what MRI is, right after. I would suggest the authors to mention which technique they refer to (e.g: …assessed by techniques such as MRI). Is it MRI? If it is the case (MRI), write MRI and give some references where the reader can find were it is assessed. For example, Mustafa Bozdag˘ et al, in their work titled “MRI assessment of pituitary iron accumulation by using pituitary-R2 in b-thalassemia patients” (DOI: 10.1177/0284185117730099), report a threshold pituitary-R2 value for thalassemia patients with hypogonadism from those with normal pituitary functions. Another work that can be cited is the work of Anderson et al titled “Cardiovascular T2-star (T2*) magnetic resonance for the early diagnosis of myocardial iron overload” (doi:10.1053/euhj.2001.2822) were they report T2* values for normal organs such as Heart, liver and  spleen mentioned in this review. Or other reference for normal T2* value threshold (doi: 10.1097/RLI.0b013e3181862413).

It was suggested in the first review, that the authors discuss the biological fate of the iron-chelator complexes. This information will be important as it will display the importance of using these chelators for the elimination of the overload-iron (NTBI) which is not effectively excreted by the body. One reference that can be used for this matter is another review publish by Galanello and Origa titled “Beta-thalassemia” were they mentioned that chelators which allow iron excretion through the urine. Using DFO specifically, iron is excreted both in feces and in urine (https://doi.org/10.1186/1750-1172-5-11)

No change was made in the conclusion as recommended in the first review.

I find it unfortunate that the authors found the original critique to be abusive. That was not the intent but it seems that the perceived negativity prevailed in the authors’ absence of a response to several serious issues throughout the article. The authors are not mindful of how they format their review. While the content is there, the overall organization is not particularly strong.  

Typing errors

P1L34.- “in homogenous” instead of “inhomogenous”

P3L94.- “treatment for thalassemia was” instead of “treatmentfor thalassemiawas”

P3L99.- “…age of 20” instead of “age of 20y”

P4L125.­- “In thalasaemic patients” instead of “Inthalasaemic patients”

P4L126.­- “such that a splenectomy” instead of “such that asplenectomy”

P4L128.­-“ four thalassaemia patients” instead of “fourthalassaemia patients”

P1L33-35.- In the revise manuscript, the authors did not review whether there is a threshold for a toxic iron concentration as recommended but mention that their effects on water protons as they diffuse in the magnetically inhomogenous environment induced by iron deposition are assessed. The question is assessed by what technic? I am guessing that they were probably thinking MRI, as they describe what MRI is, right after. I would suggest the authors to mention which technique they refer to (e.g: …assessed by techniques such as MRI). Is it MRI? If it is the case (MRI), write MRI and give some references where the reader can find were it is assessed. For example, Mustafa Bozdag˘ et al, in their work titled “MRI assessment of pituitary iron accumulation by using pituitary-R2 in b-thalassemia patients” (DOI: 10.1177/0284185117730099), report a threshold pituitary-R2 value for thalassemia patients with hypogonadism from those with normal pituitary functions. Another work that can be cited is the work of Anderson et al titled “Cardiovascular T2-star (T2*) magnetic resonance for the early diagnosis of myocardial iron overload” (doi:10.1053/euhj.2001.2822) were they report T2* values for normal organs such as Heart, liver and  spleen mentioned in this review. Or other reference for normal T2* value threshold (doi: 10.1097/RLI.0b013e3181862413).

It was suggested in the first review, that the authors discuss the biological fate of the iron-chelator complexes. This information will be important as it will display the importance of using these chelators for the elimination of the overload-iron (NTBI) which is not effectively excreted by the body. One reference that can be used for this matter is another review publish by Galanello and Origa titled “Beta-thalassemia” were they mentioned that chelators which allow iron excretion through the urine. Using DFO specifically, iron is excreted both in feces and in urine (https://doi.org/10.1186/1750-1172-5-11)

No change was made in the conclusion as recommended in the first review.

I find it unfortunate that the authors found the original critique to be abusive. That was not the intent but it seems that the perceived negativity prevailed in the authors’ absence of a response to several serious issues throughout the article. The authors are not mindful of how they format their review. While the content is there, the overall organization is not particularly strong.  

Typing errors

P1L34.- “in homogenous” instead of “inhomogenous”

P3L94.- “treatment for thalassemia was” instead of “treatmentfor thalassemiawas”

P3L99.- “…age of 20” instead of “age of 20y”

P4L125.­- “In thalasaemic patients” instead of “Inthalasaemic patients”

P4L126.­- “such that a splenectomy” instead of “such that asplenectomy”

P4L128.­-“ four thalassaemia patients” instead of “fourthalassaemia patients”

Author Response

In this review we ae discussing clinical conditions where iron chelators have been used to reduce the iron content in the pituitary of thalasaaemia patients and various brain regions of patients with neurodegenerative diseases. Such  increases  before chelation are in the order of 2-5 fold  that of control values.  In contrast in the liver and spleen of thalassaemia patient the iron content increase in excess of 20 to 50 fold.

Exactly what the  ‘threshold for a toxic iron concentration’ is undefinable.

Clearly all of the studies discussed are by MRI. We have now inserted, as assessed by MRI, in line 130. All subsequent papers already state that MRI was used for assessment of iron content.  I note his comments with respect to the quantitation of iron. In the paper by Mustafa Bozdag pituitary iron content was quantitated by MRI in thalassamic patients +/- hypogonadism and controls and  showed a 2 fold increase in pituitary R2 (Hz) in the first group by comparison to controls. However in this publication no chelation studies were reported.  It should be noted that  there is still a large gap between the MRI observations and interpretation of these changes in terms of the microscopic, molecular, and concentration of the iron. Although MRI  techniques are now sufficiently developed, the interpretation of the data in the biological context remains rudimentary.  A better understanding of the nature of the MRI signals is still required

We disagree that the fate of iron-chelator complexes should be discussed. Low dose iron chelators are utilised to remove iron from the different brain regions in neurodegenerative diseases, and in our unpublished studies of iron excretion after deferiprone administration to PD patients very low amounts of the urinary iron complexes were assayed.

Again the reviewer mentions NTBI, which is not present in neurodegenerative diseases, the iron homeostatic measurements being within normal ranges.

We do not agree with the comments by this reviewer, that the conclusion should be changed. Furthermore to state  that we are showing negativity as we do not concur with his comments  is totally untrue.  This was an invited review, and we would have thought that objections by this reviewer would have been more positive.

Furthermore there appears to have been a problem with the transmission of the paper to the Journal since on our copy the pagination was correct.